# VIDEOITG: MULTIMODAL VIDEO UNDERSTANDING WITH INSTRUCTED TEMPORAL GROUNDING

## ABSTRACT

While Video Large Language Models (Video-LLMs) have shown significant potential in multimodal understanding and reasoning tasks, how to efficiently select the most informative frames from videos remains a critical challenge. Existing methods attempt to optimize frame sampling by reducing inter-frame redundancy or employing unsupervised event localization. However, these approaches often fall short in handling complex instruction-following tasks and scenarios that demand precise temporal modeling, resulting in limited performance in both semantic alignment and temporal reasoning. To address the above challenges, we introduce *Instructed Temporal Grounding for Videos* (**VideoITG**), a framework aiming to adaptively customize frame sampling strategies based on user instructions. Specifically, we design the VidThinker pipeline, which automates annotation by generating instruction-conditioned captions, retrieving relevant video segments, and selecting key frames to enable efficient supervision. Using VidThinker, we build the VideoITG-40K dataset with 40K videos and 500K temporal grounding annotations. Our plug-and-play VideoITG model leverages Video-LLMs' visual-language alignment and reasoning for discriminative frame selection. VideoITG consistently boosts the performance on multiple multimodal video understanding benchmarks, demonstrating its effectiveness and potential.

## 1 INTRODUCTION

The rapid advancement of Video Large Language Models (Video-LLMs) has opened new frontiers in video understanding, advancing complex tasks such as captioning (Chen et al., 2024c; Chai et al., 2025; Zhou et al., 2024b; Islam et al., 2024; Chen et al., 2024d; Wang et al., 2024b), visual question answering (Fu et al., 2024a; Zhou et al., 2024a; Mangalam et al., 2024; Li et al., 2024b; Chen et al., 2024a; Xiao et al., 2021; Pătrăucean et al., 2023), and embodied-agent applications (Brohan et al., 2023; Kim et al., 2024; Fu et al., 2024b; Liu et al., 2024a; Chen et al., 2024b; 2025b). However, these models face challenges when handling long videos due to high memory and computational demands. To mitigate this, existing approaches often adopt uniform frame sampling, a simple but naive strategy that frequently misses key frames for accurate video understanding, resulting in suboptimal performance.

To address these limitations, researchers have explored various strategies. One family of approaches focuses on reducing redundant spatiotemporal information by fusing or pruning overlapped content across frames, as can be seen in works employing pooling rules, similarity thresholds, or clustering to retain only essential frames (Xu et al., 2024a; Shen et al., 2024; Zhang et al., 2024a; Li et al., 2024a; Zhang et al., 2024c). Another stream of strategies extends the length of model sequence to incorporate more tokens (Wang et al., 2024c; Team et al., 2023), enabling longer temporal dependencies, despite the high computational cost, and risking information dilution. Alternative methods utilize question-guided feature extraction or language queries to identify relevant segments (Li et al., 2024c; Yu et al., 2023). SeViLA (Yu et al., 2023) processes frames independently using BLIP-2 (Li et al., 2023) before selecting keyframes, which serve as the input for subsequent video understanding tasks. However, the lack of temporal modeling capability limits its performance in tasks requiring multi-temporal cues.

Despite advances in compressing or extending the context for Video-LLMs, a performance gap persists between long and short videos due to limited training data for long-video content. When humans analyze long videos, they naturally employ a step-by-step approach: skimming the overall

Figure 1: Overview of the VidThinker annotation pipeline for VideoITG. The pipeline consists of three stages that fully leverage the provided instructions: (1) segment-level clip captioning; (2) instruction-guided relevant clip retrieval; (3) fine-grained frame-level localization.

context, locating question-relevant clues, and then focusing on specific segments. Drawing inspiration from such a process, we propose **I**nstructed **T**emporal **G**rounding for **Video**s (**VideoITG**), which integrates user instructions into frame selection. While general temporal video grounding (Wang et al., 2024a; Qian et al., 2024; Lei et al., 2021) emphasizes event localization within videos based on single temporal clue and descriptive language queries, VideoITG introduces a user-instruction-driven approach, customizing frame selection strategies to align with specific task requirements. Compared to existing frame selection frameworks (Yu et al., 2023; 2025; Han et al., 2025; Meng et al., 2022; Wang et al., 2022), VideoITG can effectively handle multiple temporal clues for various tasks: localizing temporal cues from multiple clips to understand temporal relationships, employing event localization and uniform frame sampling to detect speed variations, and conducting diverse types of samplings to cover all videos for content captioning or existence judgment, *etc*.

To support VideoITG, we construct a comprehensive dataset via the automated annotation pipeline *VidThinker*, which includes instruction-guided clip captioning, retrieval, and frame localization to ensure high-quality, task-aligned annotations (see Fig. 1). Inspired by human reasoning, our pipeline uses GPT-4o (OpenAI, 2024) for detailed descriptions and a "Needle-In-A-Haystack" approach for instruction-guided retrieval. To achieve precise temporal grounding, instructions are categorized into four types: **semantic-only** for appearance-based questions, **motion-only** for dynamic cues, **semantic & motion** for joint reasoning, and **non-clues** for maximizing visual diversity.

The resulting VideoITG-40K dataset contains 40K videos with varying durations (30s - 3mins) and 500K instruction-guided annotations, significantly surpassing previous temporal grounding datasets in both scale and quality of instruction-guided frame selection. Building on this foundation, we present a family of VideoITG models that leverage text generation, anchor-based classification with causal attention, and pooling-based classification with full attention to enhance instructed temporal grounding and advance Video-LLM capabilities. In summary, our contributions are threefold:

- **VideoITG-40K dataset.** We define the tasks of VideoITG and develop an automated data annotation pipeline, namely *VidThinker*, to generate a large-scale dataset, namely VideoITG-40K, with 40K videos and 500K instruction-dependent annotations, allowing precise frame identification and effective video understanding.

- **VideoITG models.** We introduce a family of VideoITG models with varying attention and decoding strategies, designed to improve instruction-guided temporal grounding based on insights from the VideoITG-40K dataset.

- **Consistent improvement.** Our approach achieves consistent performance improvements on various multimodal video understanding benchmarks. By integrating VideoITG, we achieve improvements of **9.0%** on CG-Bench, **8.6%** on MLVU, **4.0%** on Video-MME, and **3.6%** on LongVideoBench for the InternVL2.5-8B model, showing the effectiveness of our framework.

## 2 RELATED WORK

**Video large language models.** Recent advances in Video-LLMs address the temporal and spatial complexity of long videos through several strategies. Visual feature compression (Liu et al., 2025b; Zohar et al., 2024; Ye et al., 2024; Wang et al., 2024e; Liu et al., 2025a) is achieved by modules like Q-Former (Song et al., 2024) and Perceiver Resampler (Zohar et al., 2024), which merge frame features into fixed queries. Spatial pooling (Maaz et al., 2024; Xu et al., 2024b;a) helps preserve long-range temporal information efficiently. Some models extend sequence length for longer inputs (Zhang et al., 2024d; Wang et al., 2024c; Team et al., 2023; Chen et al., 2025a), but this often increases computational cost (Wei & Chen, 2024; Shu et al., 2025). To reduce redundancy, similarity-based frame filtering is used (Jin et al., 2024; Shen et al., 2024), though fixed thresholds may miss real-world diversity. Some methods also incorporate additional components for frame-by-frame selection; however, they do not sufficiently model temporal relationships (Huang et al., 2025; Wang et al., 2024d; Buch et al., 2022). In contrast, our VideoITG leverages instructed temporal grounding, automated annotation, and a plug-and-play design to align sampling with user instructions, achieving superior performance and scalability on multimodal video understanding benchmarks.

**Video temporal grounding.** Video Temporal Grounding (Ren et al., 2024; Wang et al., 2024a; Qian et al., 2024; Di & Xie, 2024) is a common task in video understanding that associates specific video moments with their corresponding timestamps, while Temporal Localization focuses on accurately identifying these moments within untrimmed videos (Liu et al., 2024b; Anne Hendricks et al., 2017; Li et al., 2024d). Current Video-LLMs(Shen et al., 2024; Wang et al., 2024a; Huang et al., 2024) have begun to leverage temporal grounding for frame selection by linking video content with temporal cues; however, existing methods (Huang et al., 2025; Yu et al., 2023; 2025) mostly focus on single-time retrieval, which take descriptive annotations as input, limiting their generality and robustness in handling diverse real-world scenarios. Recognizing these limitations, we propose the VideoITG task, which introduces a customized sampling approach aligned with user instructions to improve the effectiveness of frame selection for a broad range of video understanding tasks.

## 3 VIDEOITG-40K: DATASET CONSTRUCTION

We introduce *VidThinker* (Sec. 3.1), an automated annotation pipeline with three stages—clip captioning, retrieval, and frame localization for instruction-based video annotation. We further describe our fine-grained instruction taxonomy and frame selection strategies (Sec. 3.2) to align annotations with QA tasks. Finally, we apply *VidThinker* to construct the VideoITG-40K dataset and report its statistics (Sec. 3.3).

### 3.1 VIDTHINKER: AUTOMATED ANNOTATION PIPELINE

Instruction-driven temporal localization in long videos typically involves three steps: parsing the instruction to extract key information, narrowing down the video to a coarse temporal window, and fine-grained reasoning to locate the target event. Inspired by this process, we propose *VidThinker*, an automated annotation pipeline that mimics human reasoning for instruction-guided temporal localization. *VidThinker* enables fully automated, high-quality, and interpretable video annotations without manual labeling.

*VidThinker* decomposes the annotation process into three interdependent reasoning steps: i) Instructed Clip Captioning, ii) Instructed Clip Retrieval, and iii) Instructed Frame Localization. It progressively narrows the search space and enriches semantic alignment with the instruction.

**i) Instructed Clip Captioning:** The video $v$ is uniformly divided into short clips (5 seconds each), denoted as $\{v_i\}_{i=0}^n$. For each segment, we employ LLM to extract salient phrases that capture the core information needed to fulfill the instruction. For example, given the question ($q =$ 'What does the man playing the drums do with his feet as he plays the drum?') and the answer ($a =$ 'moves his feet'), the system distills the essential action phrase: $k =$ 'The man playing the drums moves his feet and hits the drums with his hands.' We then input the extracted phrases alongside raw video clips into the MLLM to generate clip-level descriptions $\{c_i\}_{i=0}^n$ in a recurrent manner. The extracted phrases serve as reference cues to guide the model's attention towards salient elements within each clip. However, the MLLM strictly adheres to visual evidence and it only incorporates information from the extracted

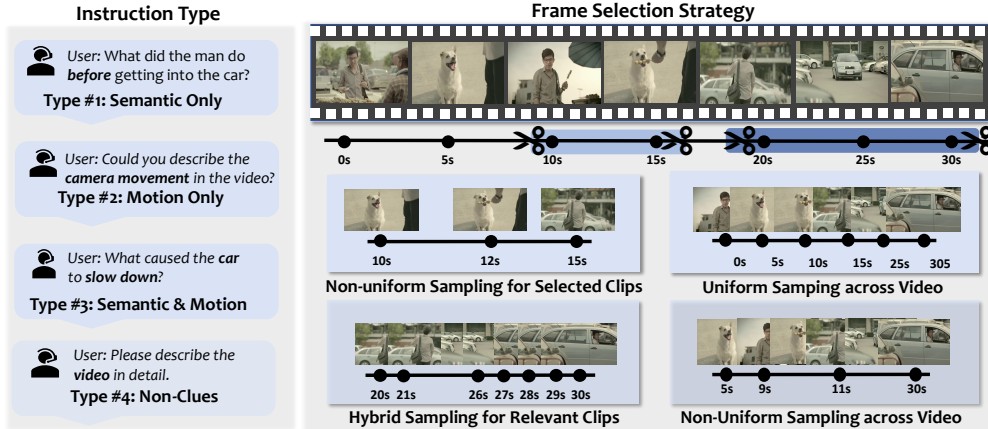

Figure 2: Illustration of four instruction types and their corresponding frame selection strategies in VidThinker. For semantic-focused instructions, the system selects diverse frames capturing key visual clues. For motion-focused instructions, frames are uniformly sampled to capture dynamic changes. When both semantic and motion cues are required, a hybrid sampling strategy is applied. For vague or open-ended instructions, the system samples a minimal yet diverse set of frames across the video for holistic coverage.

phrases when it is explicitly observable in the current clip. This ensures that the system will not hallucinate or infer content solely based on the extracted phrases, maintaining descriptions grounded in visual content. The process can be formulated as follows:

$$k = \text{LLM}(q, a), \quad c_i = \text{MLLM}(k, v_i). \tag{1}$$

Conditioning on these instruction- and answer-derived cues, we ensure that the annotation of each segment is relevant and informative, thus facilitating precise instructed temporal grounding.

**ii) Instructed Clip Retrieval:** The generated clip descriptions $\{c_i\}_{i=0}^n$ are organized sequentially and evaluated by an LLM for the relevance to the QA pairs. Instead of simply assigning binary relevance scores, the LLM is instructed to perform chain-of-thought reasoning, explicitly considering both keyword matches and temporal relationships to directly output the indexes of relevant clips:

$$\mathcal{I}_{\text{rel-clip}} = \text{LLM}(\{c_i\}_{i=0}^n, q, a). \tag{2}$$

The chain-of-thought prompting requires the model to justify its selections based on both semantic and temporal cues, rather than relying solely on trivial keyword matching. This automation significantly improves the efficiency and the interpretability of relevant segment selection.

**iii) Instructed Frame Localization:** After coarse localization of video segment, *VidThinker* further refines the annotation by selecting key frames according to the instruction type. For each frame within the candidate segment, we prompt a large language model (LLM) to perform a binary classification task: given the QA pair and a single frame, the LLM determines whether the frame is relevant (yes) or not (no) to the instruction. Formally, for each frame $f_i$ in the candidate segment, the LLM is prompted as follows:

$$y_i = \text{LLM}(f_i, q, a), \quad \text{where} \quad y_i \in \{\text{yes}, \text{no}\}, \tag{3}$$

where $y_i$ indicates whether frame $f_i$ is relevant to the QA. Only frames with positive responses ($y_i = \text{yes}$) are retained as the final temporal grounding results. This instruction-guided filtering allows *VidThinker* to achieve high precision in identifying the most informative frames for instructions.

Leveraging the reasoning capabilities of MLLMs, *VidThinker* transforms video QA annotation into a fully automated, scalable, and cognitively inspired process. This approach not only reduces manual effort and variability, but also ensures high-quality, interpretable annotations suitable for training next-generation video understanding models.

## 3.2 FINE-GRAINED GROUNDING INSTRUCTION

We adopt fine-grained frame selection strategies tailored to each instruction type, ensuring that the visual evidence matches the reasoning needs of each QA task. Since different instructions demand

Table 1: Comparison of dataset statistics for temporal grounding and highlight detection datasets.

| Dataset | # Videos | # Queries | Avg. Duration | Instructed? |
|---|---|---|---|---|
| DiDeMo (Anne Hendricks et al., 2017) | 10.6K | 41.2K | 29s | No |
| QuerYD (Oncescu et al., 2021) | 2.6K | 32K | **278s** | No |
| HiREST (Zala et al., 2023) | 3.4K | 8.6K | 263s | No |
| Charades-STA (Gao et al., 2017) | 6.7K | 16.1K | 30s | No |
| QVHighlights (Lei et al., 2021) | 10.2K | 10.3K | 150s | No |
| VideoITG-40K | **40K** | **500K** | 120s | **Yes** |

varying visual understanding, such as static semantics, dynamic motion, or both, we categorize instructions into four types and apply different sampling methods.

- **Semantic only**: These instructions focus on semantic content such as people, scenes, or objects. Following relevant segment localization, the system selects diverse frames that capture representative visual clues to ensure comprehensive semantic coverage. For example: *"What did the man do before getting into the car?"* The *VidThinker* needs to select frames that clearly show the man's clothing and the guitar.

- **Motion only**: These instructions emphasize dynamic actions, such as movement type, speed, or direction. The frames are sampled at a fixed rate within the localized segment to accurately capture the progression of motion. For example: *"How does the person jump off the diving board?"* The *VidThinker* needs to select frames covering the sequence from takeoff, mid-air, to water entry.

- **Semantic & Motion**: These instructions require both semantic and motion understanding. The system applies fixed-rate sampling within motion-relevant regions while ensuring the preservation of semantically informative frames, balancing both needs. For example: *"Could you describe the camera movement in the video?"* The *VidThinker* needs to select frames showing hand drumming and foot movement simultaneously.

- **Non Clues**: These instructions are open-ended or vague without clear semantic or motion focus, aiming to maximize visual diversity for holistic understanding. In these cases, the system selects a small yet diverse set of frames across the entire video to maximize visual information coverage while minimizing redundancy. For example: *"Please describe the video in detail."* The *VidThinker* selects representative frames from the beginning, middle, and end of the video.

## 3.3 DATASET STATISTICS

Leveraging our proposed *VidThinker* pipeline, we construct the VideoITG-40K dataset, which is sourced from the LLaVA-Video dataset (Zhang et al., 2024e). VideoITG-40K achieves an unprecedented scale, comprising 40,000 videos and 500,000 annotations tailored specifically for instruction-guided temporal grounding. The entire annotation process is automatically carried out by *VidThinker*, ensuring high efficiency, consistency, and alignment with diverse instruction types.

VideoITG-40K contains videos of varying duration, averaging 120 seconds, and is uniformly sampled across the timelength of 30-60s, 1-2mins, and 2-3mins. Each video is comprehensively annotated with 10–15 QA pairs, including both multiple-choice and open-ended questions. As shown in Table 1, VideoITG-40K significantly surpasses existing datasets in volume, with nearly four times the number of videos compared to DiDeMo (Anne Hendricks et al., 2017) (10.6K) and QVHighlights (Lei et al., 2021) (10.2K), and far exceeding others like QuerYD (Oncescu et al., 2021) (2.6K) and HiREST (Zala et al., 2023) (3.4K). Unlike prior datasets that primarily focus on descriptive text queries for video understanding, VideoITG-40K distinguishes itself through its instruction-guided approach, enabling models to locate relevant video content based on specific user queries.

## 4 VIDEOITG: MODEL DESIGN

In this section, we explore how to utilize our VideoITG-40K dataset to train the model for the **Instructed Temporal Grounding** task, aiming to optimize video frame selection and enhance the performance of Video-LLMs. Our framework, as shown in Fig. 3, consists of three main components: a vision encoder (*i.e.*, VIT) for extracting text-aligned visual features $F$, a VideoITG model for instruction-guided frame selection $\mathcal{I}_{\text{rel}}$, and a VideoLLM for generating answers $a$ based on the

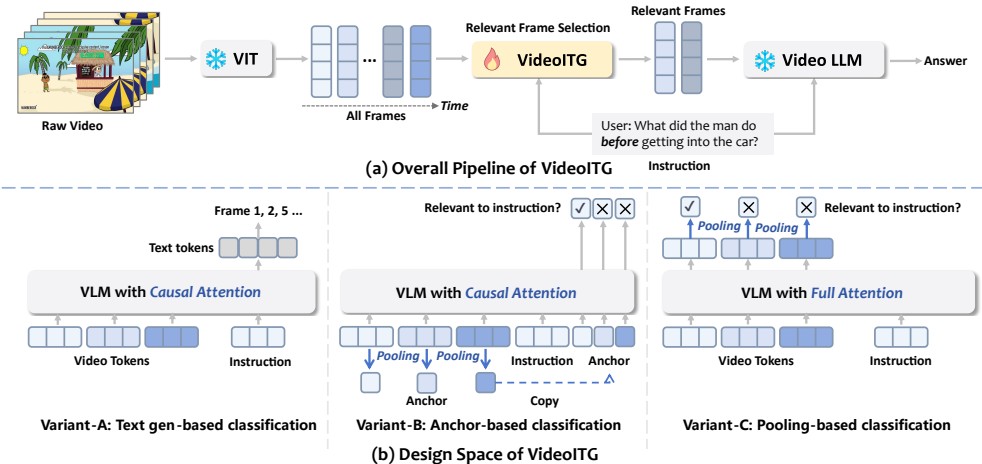

**(a) Overall Pipeline of VideoITG**

**(b) Design Space of VideoITG**

Figure 3: **VideoITG model design**: (A) Text generation aligns video and language tokens for sequential predictions. (B) Classification with causal attention utilizes anchor tokens for temporal cue management. (C) Classification with full attention facilitates interaction across visual and text tokens without anchors.

selected frames $F_{\mathcal{I}_{rel}}$ and the question $q$. The process can be described as folllows:

$$F = \text{VIT}(v), \quad \mathcal{I}_{rel} = \text{VideoITG}(F, q), \quad a = \text{VideoLLM}(F_{\mathcal{I}_{rel}}, q). \quad (4)$$

The VideoITG model is designed in a plug-and-play fashion, which is focused on the following aspects: (1) to which extent the Video-LLMs capitalize on the alignment between video and language tokens, as well as their ability to follow instructions; (2) whether the model has sufficient contextual encoding capabilities to handle and analyze multiple temporal cues. With the above considerations, we develop three model variants: text generation-based classification, anchor-based classification, and pooling-based classification, as illustrated in Fig. 3 (b).

**Variant A: text-generation-based classification.** As shown on the left side of Fig. 3 (b), we start by discussing the text generation design, where Instructed Temporal Grounding is framed as a next-token prediction task, producing text tokens as output. This approach aligns with the current training paradigm of Video-LLMs, optimizing the use of Video-LLM for vision-language alignment and instruction following. Previous works, such as Timechat (Ren et al., 2024) and Grounded-VideoLLM (Wang et al., 2024a), use this paradigm to tackle time-sensitive tasks.

**Variant B: anchor-based classification.** Another model design adopts a discriminative paradigm by classifying the visual tokens corresponding to each video frame, as shown in the middle of Fig. 3 (b). We initialize the model with a Video-LLM and retain the causal attention mask to remain consistent with the original Video-LLM design, thereby preserving its pre-training capabilities. However, the causal attention mask prevents visual tokens from accessing the instruction in advance, and earlier frame features are unable to access subsequent frame features, limiting the model's ability to handle multiple temporal cues. To address this limitation, we introduce an *anchor token* after the instruction. For a video frame at time index $t$, we compute an anchor token $A^t$ via global average pooling over all spatial positions, as formulated below:

$$A^t = \frac{1}{M} \sum_{i,j} F_{ij}^t, \quad \forall t \in [1, \cdots, T], \quad (5)$$

where $F_{ij}^t$ represents the visual feature extracted at 2D grid position $(i, j)$ of the $t$-th frame, and $M$ is the total number of patches within each frame. For a video with $T$ frames, we compute $T$ anchor tokens in total: $\{A^t\}_{t=1}^T$.

**Variant C: pooling-based classification.** As discussed above, the presence of a causal attention mask makes it difficult to directly supervise the classification of visual tokens for each frame. Therefore, we propose to remove the causal attention mask, allowing visual tokens and text instruction tokens to interact through full attention across the sequence. Following this idea, we perform average pooling and classification on the visual tokens of each frame without establishing separate anchor tokens. The overall process is illustrated on the right of Fig. 3 (b).

Table 2: Performance comparison of VideoITG integrated with different Video-LLMs, varying in both the size of the answering LLM and the number of sampled frames. "UNI-$k$" denotes UNIform sampling of $k$ frames, while "ITG-$k$" refers to selecting the Top $k$ frames based on relevance scores generated by our proposed VideoITG.

| LMM | Selection | LongVideoBench | MLVU | VideoMME | | | CG-Bench | Avg. |
|---|---|---|---|---|---|---|---|---|
| | | 8min | 12min | S (2 min) | M (10 min) | L (40 min) | 27min | |
| LLaVA-Video-7B | UNI-32 | 58.7 | 66.8 | 76.3 | 60.3 | 52.7 | 35.8 | 58.4 |
| | ITG-32 | 61.6 (+2.9) | 74.6 (+7.8) | 77.3 (+1.0) | 65.9 (+5.6) | 55.2 (+2.5) | 42.8 (+7.0) | 62.9 (+4.5) |
| LLaVA-Video-7B | UNI-64 | 59.9 | 70.2 | 75.8 | 63.0 | 54.7 | 36.9 | 60.1 |
| | ITG-64 | 60.9 (+1.0) | 76.3 (+6.1) | 76.1 (+0.3) | 66.0 (+3.0) | 57.0 (+2.3) | 42.9 (+6.0) | 63.2 (+3.1) |
| InternVL2.5-8B | UNI-32 | 58.3 | 66.4 | 75.1 | 61.7 | 53.1 | 37.7 | 58.7 |
| | ITG-32 | 61.9 (+3.6) | 75.0 (+8.6) | 78.0 (+2.9) | 67.1 (+5.4) | 56.9 (+3.8) | 46.7 (+9.0) | 64.3 (+5.6) |
| InternVL2.5-26B | UNI-32 | 55.6 | 71.3 | 78.1 | 67.1 | 56.9 | 40.6 | 61.6 |
| | ITG-32 | 63.0 (+7.4) | 78.9 (+7.6) | 80.8 (+2.7) | 69.0 (+1.9) | 59.9 (+3.0) | 48.7 (+8.1) | 66.7 (+5.1) |
| Eagle2.5-8B | UNI-32 | 63.0 | 67.8 | 78.8 | 64.1 | 55.9 | 41.2 | 61.8 |
| | ITG-32 | 66.8 (+3.8) | 76.5 (+8.7) | 80.0 (+1.2) | 67.8 (+3.7) | 60.3 (+4.4) | 49.0 (+7.8) | 66.7 (+4.9) |

## 5 EXPERIMENTS

### 5.1 IMPLEMENTATION DETAILS

We follow the training approach of LLaVA-Video (Zhang et al., 2024e), using the pretrained model as the initialization for our VideoITG model's pre-training. We employ SigLIP (Zhai et al., 2023) as the vision encoder and Qwen2 (Wang et al., 2024c) as the language model. Initially, we train the MLP projector on image caption datasets with a batch size of 256 and a learning rate of $1 \times 10^{-3}$. Then, we fine-tune all model parameters on the LLaVA-OV-SI (Li et al., 2024a) and LLaVA-Video datasets. During this stage, the video frame sampling rate is set to 64, and the LLM's maximum sequence length is set to 16K. We then train the VideoITG model on the proposed VideoITG-40K dataset, adjusting the video sampling rate to 1 fps.

Throughout training and inference, we employ a dynamic token spatial size strategy (Liu et al., 2025b). Across all stages, the LLM's learning rate is $2 \times 10^{-5}$, and in the final stage, the learning rate for the classification head is $2 \times 10^{-4}$. To fairly compare with other leading video LMMs, we primarily use results from their original papers. When results are unavailable, we integrate the models into LMMs-Eval (Zhang et al., 2024b) and assess them under consistent settings. Due to context length constraints, we support up to 512 video frames as input (with 16 visual tokens per frame) for the VideoITG model, from which we select the top 32 frames based on their scores by default.

### 5.2 MAIN RESULTS

In Table 2, we integrate our VideoITG model with various Video-LLMs to examine how different frame sampling strategies and the number of sampled frames influence answer quality. As can be seen, VideoITG's frame selection strategy significantly outperforms uniform sampling across both 32-frame and 64-frame settings. This demonstrates that uniform sampling indeed constrains the extraction of informative content within the limited frame budget.

We evaluate Video-LLM models of different sizes and observe that integrating VideoITG yields substantial improvements, even for larger models. For InternVL2.5-26B, our approach boosts performance by 7.4% on LongVideoBench, 7.6% on MLVU, and 9.0% on CG-Bench. Notably, InternVL2.5-8B with VideoITG achieves an average of 64.3%, surpassing the InternVL2.5-26B baseline (61.6%). On longer video tasks, InternVL2.5-8B with VideoITG attains 46.7% on CG-Bench, outperforming the 26B baseline (40.6%). These results demonstrate that effective frame selection can yield greater gains than increasing model size, especially for long-video understanding.

### 5.3 ABLATION ON VIDEOITG DESIGN CHOICES

Table 3 presents a comprehensive analysis on the design of our VideoITG framework, directly supporting our key contributions.

Table 3: Empirical studies on the VideoITG-40k dataset and VideoITG model design. We adopt Variant-C for subsequent experiments. "No Images" and "No Videos" indicate that image-text data (LAION-CC-SBU-558K & LLaVA-OV-SI) or video data (LLaVA-Video-178K) are excluded from pre-training, respectively.

| Abaltion | Experiment | Videomme | | | MLVU | LongVideoBench |
| | | Short (%) ↑ | Medium (%) ↑ | Long (%) ↑ | (%) ↑ | (%) ↑ |
|---|---|---|---|---|---|---|
| Architecture | Variant-A-7B | 51.0 | 44.8 | 44.4 | 45.7 | 56.8 |
| | Variant-B-7B | 77.9 | 66.0 | 56.2 | 74.6 | 61.3 |
| | Variant-C-7B | **78.0** | **67.1** | 56.9 | **75.0** | **61.9** |
| | Variant-C-3B | 77.1 | 64.8 | 56.0 | 74.5 | 61.5 |
| Dataset Construction | No Clip Captioning | 77.5 | 63.1 | 53.4 | 73.2 | 61.7 |
| | No Frame Localization | 77.6 | 65.8 | 56.8 | 74.1 | 61.5 |
| Pre-training Data | No Videos | 77.2 | 64.9 | **57.4** | 74.5 | 61.6 |
| | No Images & Videos | 76.6 | 63.0 | 54.4 | 69.1 | 58.6 |

**Architecture.** First, we compare the three variants of our model architecture. We observe that Variant A, which is based on the text generation paradigm, performs the worst. One possible reason is that text generation models trained with the next-token prediction paradigm suffer from sparse supervision due to teacher forcing, where previous frame selections influence subsequent ones, making the training process less efficient compared to discriminative classification models. We find that Variant C with full-attention outperforms Variant B with causal attention. This improvement may be attributed to full-attention's larger receptive field, which enables global temporal relationship modeling and allows all tokens to access the textual query simultaneously.

**Dataset.** We analyze our data annotation strategies to demonstrate the effectiveness of our *Vid-Thinker* annotation pipeline. Ablation studies show that the performance degrades when Instructed Clip Captioning are removed, with accuracy dropping from 56.9% to 53.4% on Videomme Long videos and from 75.0% to 73.2% on MLVU. This demonstrates that ensuring information diversity is crucial for maintaining comprehensive feature representation of videos. Similarly, removing Instructed Frame Localization decreases performance, particularly on Videomme Medium videos (from 67.1% to 65.8%). These results confirm that both stages are essential for optimal model performance and validate our data construction approach of the VideoITG-40K dataset.

**Pre-training.** Finally, we investigate the impact of vision-language alignment pre-training on model performance. Our experiments reveal that removing video pre-training causes only modest performance changes across benchmarks, with slight increases on Videomme Long videos. This suggests that the benefits of video data for instructed temporal grounding tasks primarily stem from effective visual context length, yet this impact is relatively minor compared to vision-language alignment. This observation is further validated if we eliminate both image and video pre-training data, starting from a text-only large language model, where performance drops dramatically, with accuracy decreasing from 75.0% to 69.1% on MLVU and from 61.9% to 58.6% on LongVideoBench. This substantial degradation underscores that robust vision-language alignment is crucial to effective VideoITG training.

## 5.4 ABLATION ON SELECTION METHODS

Table 4 presents a comprehensive comparison of various frame selection methods. Baseline approaches like SigLIP and InternVL2.5-8B achieve average scores of 64.0 and 65.2, respectively, but are limited by weaker instruction-following and temporal modeling capabilities. Recent methods such as AKS (Tang et al., 2025) and QuoTA (Luo et al., 2025) introduce more advanced selection strategies, reaching average scores of 65.3 and 65.6; however, they still fail to fully leverage temporal cues in videos. Q-Frame (Zhang et al., 2025) attempts multi-resolution scaling, but its overall performance (61.7) remains inferior. Considering results across the three major VLMs, LLaVA-Video, InternVL2.5, and Qwen2VL, VideoITG consistently achieves the best overall performance. On each VLMs, VideoITG-8B attains the highest or near-highest scores, demonstrating its robust generalization and superior capability in instruction following and temporal modeling. By jointly modeling instruction-following and temporal relationships in both data annotation and model design, VideoITG demonstrates superior video understanding, especially in complex and long-form scenarios.

Table 4: Results with different scoring LMMs. In the second row, we follow Huang et al. (2025) and use a standalone VLM to assess the relevance between the question and each frame, and select the top 32 frames with the highest probability of "Yes" as the output.

| Selection Methods | Answering LMM | Frames | LongVideoBench | MLVU | VideoMME | Avg. |
|---|---|---|---|---|---|---|
| SigLIP (Zhai et al., 2023) | InternVL2.5-8B | 32 | 60.4 | 69.3 | 62.4 | 64.0 |
| InternVL2.5-8B (Huang et al., 2025) | InternVL2.5-8B | 32 | 60.7 | 70.3 | 64.7 | 65.2 |
| VideoITG-8B | InternVL2.5-8B | 32 | **61.9** | **75.0** | **67.3** | **68.1** |
| AKS (Tang et al., 2025) | LLaVA-Video-7B | 64 | **62.7** | - | 65.3 | - |
| QuoTA (Luo et al., 2025) | LLaVA-Video-7B | 64 | 59.0 | 71.9 | 65.9 | 65.6 |
| VideoITG-8B | LLaVA-Video-7B | 64 | 60.9 | **76.3** | **66.4** | **67.9** |
| Q-Frame (Zhang et al., 2025) | Qwen2VL | 8+16+32 | 58.4 | 65.4 | 58.3 | 60.7 |
| VideoITG-8B | Qwen2VL | 8+16+32 | **58.6** | **66.6** | **59.8** | **61.7** |

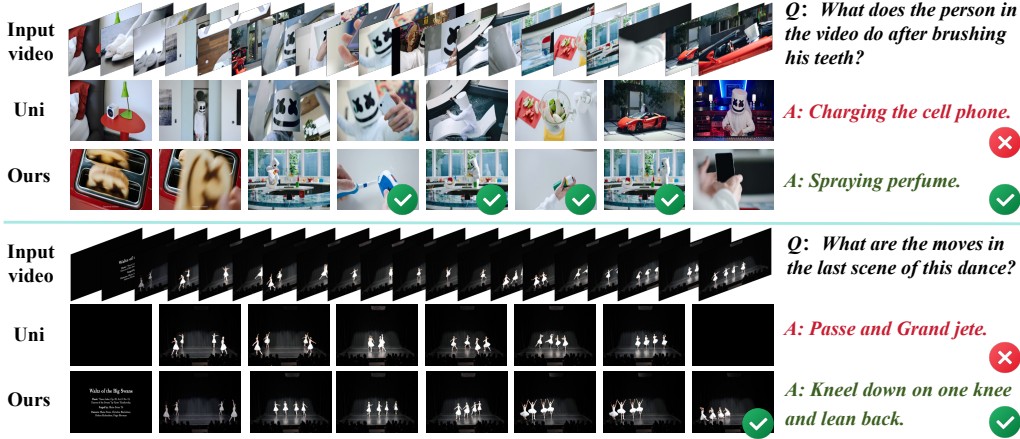

Figure 4: Two examples of how different sampling strategies impact video understanding. We mark the identified key frames that directly answer the question with green check-marks.

## 5.5 VISUALIZATION

In Fig. 4, we compare uniform sampling and VideoITG sampling of 8 frames from the VideoMME (Fu et al., 2024a) Benchmark. In the first case, VideoITG captures both brushing teeth and spraying perfume actions, enabling correct temporal ordering, while uniform sampling misses key cues. In the second case, VideoITG accurately captures rapid consecutive movements at the end, whereas uniform sampling fails to do so, leading to incomplete video understanding.

## 6 CONCLUSION

In this paper, we presented VideoITG, a novel framework for instruction-aligned frame selection in Video-LLMs. The key to our approach was the *VidThinker* pipeline, which mimics human annotation by generating detailed, instruction-guided clip descriptions, retrieving relevant segments, and performing fine-grained frame selection. Using this pipeline, we constructed the VideoITG-40K dataset with 40K videos and 500K temporal grounding annotations. Based on this resource, we developed plug-and-play VideoITG models that leverage visual-language alignment and reasoning to handle diverse temporal grounding tasks. Experiments showed that VideoITG consistently improves Video-LLMs' performance across multiple video understanding benchmarks, highlighting its effectiveness and potential for advancing instruction-driven video understanding.

**Limitations.** Our current framework consists of two separate modules during inference: VideoITG for frame selection and a standalone Video-LLM for question answering. Although we have carefully designed the frame labeling strategies, the lack of gradient optimization between these two modules during training can lead to suboptimal results. Our VideoITG framework serves as a promising starting point, while in future work we could explore reinforcement learning techniques to bridge these two modules, enabling more efficient and accurate frame selection.

## 7 ETHICS STATEMENT

Our methods are intended solely for academic and scientific purposes. We do not foresee direct harmful applications, but acknowledge that misuse could occur if applied without proper safeguards. We encourage responsible use of the research outcomes, with attention to fairness, transparency, and legal compliance. The usage of all datasets strictly complies with their respective licenses.

## 8 REPRODUCIBILITY STATEMENT

We have taken several measures to ensure the reproducibility of our work. All details of the proposed model, preprocessing steps of datasets and algorithms with full hyperparameter settings and training procedures provided are described in the main text.

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

---

**Algorithm 1** Keyframe Extraction via Bidirectional CLIP Similarity

---

**Require:** Video frame sequence `frames`, similarity thresholds $t_1$ (scene change) and $t_2$ (diversity)
**Ensure:** Selected keyframe indices `sel`
 1: Initialize `sel` with the first frame index: `sel` $\leftarrow \{0\}$
 2: Extract CLIP feature for the first frame: `prev` $\leftarrow$ clip(`frames`[0])
 3: **for** each frame $c$ in `frames[1:]` with index $i$ **do**
 4:     `curr` $\leftarrow$ clip($c$)
 5:     $s \leftarrow$ sim(`curr`, `prev`)
 6:     **if** $s < t_1$ **then**
 7:         **for** each future frame $f$ in `frames`$[i + 1 :]$ **do**
 8:             `fut` $\leftarrow$ clip($f$)
 9:             **if** sim(`curr`, `fut`) $< t_2$ **then**
10:                 Add index $i$ to `sel`
11:                 `prev` $\leftarrow$ `curr`
12:                 **break**
13:             **end if**
14:         **end for**
15:     **end if**
16: **end for**
17: **if** sim(clip(`frames`[-1]), `prev`) $< t_1$ **then**
18:     Add last frame index to `sel`
19: **end if**
20: **return** `sel`

---

## A  THE USE OF LARGE LANGUAGE MODELS (LLMs)

In this work, Large Language Models (LLMs) were employed in four main ways: (i) to aid and polish the writing for clarity and style; and (ii) to provide coding assistance, including code generation, debugging, and optimization suggestions.

Specifically, LLMs were utilized to improve the clarity, coherence, and readability of the manuscript, with particular attention given to the **Related Work**, **Method**, and **Experiments** sections. In these parts, the initial drafts were carefully reviewed and refined using LLM-powered suggestions for sentence structure, terminology, and logical flow. This process ensured that the technical content was presented in a precise and accessible manner, while maintaining consistency in academic tone and style throughout the paper.

All outputs generated by LLMs were critically reviewed, verified, and further refined by the authors. The core scientific ideas, methodology, and contributions remain entirely the authors' own. The use of LLMs was strictly limited to language enhancement and coding support, without influencing the originality or integrity of the research.

## B  INFERENCE TIME

In Table 5, we evaluated the speed of our model on a single NVIDIA A100 GPU. We employed LLaVA-Video-7B (Zhang et al., 2024e) as our answering LLM, implemented a 32-frame sampling strategy from 512 input frames in total, and generated 27 text tokens. Additionally, we leveraged KV Cache and Flash Attention (Dao et al., 2022; Dao, 2024) to enhance inference efficiency.

Our detailed analysis of computational costs reveals that processing each video sample requires a total of 6.42 seconds, with the Vision Encoder (2.92 seconds) and LLM (2.89 seconds) dominating the time consumption. These two components collectively consume 90% of the total processing time, indicating the direction for future system optimization. In contrast, our VideoITG module demonstrates remarkable efficiency, requiring only 0.61 seconds to scan 512 frames—a speed that surpasses human visual recognition and thinking capabilities.

Table 5: Computation cost of the model.

| Vision Encoder | VideoITG | LLM | Overall |
|:---:|:---:|:---:|:---:|
| 2.92s | 0.61s | 2.89s | 6.42s |

Table 6: The performance (accuracy) of SOTA methods on video benchmarks. For InternVL2.5-8B results, we report the higher results in the technical report and lmms-eval. We sample 32 frames using VideoITG for our results.

| | Open-Ended Q&A | Multi-Choice Q&A | | | | | | |
|---|---|---|---|---|---|---|---|---|
| **Model** | ActNet-QA | EgoSchema | MLVU | NExT-QA | PerceptionTest | LongVideoBench | VideoMME | MVBench |
| | test | test | m-avg | mc | val | val | wo/w-subs | val |
| *Open-source models* | | | | | | | | |
| VILA-40B (Lin et al., 2024) | 58.0 | 58.0 | - | 67.9 | 54.0 | - | 60.1/61.1 | - |
| PLLaVA-34B (Xu et al., 2024a) | 60.9 | - | - | - | - | 53.2 | - | 58.1 |
| VideoLLaMA2-7B (Cheng et al., 2024) | 50.2 | 50.5 | - | - | 49.6 | - | 45.1/46.6 | 53.4 |
| LongVA-7B (Zhang et al., 2024d) | 50.0 | - | 56.3 | 68.3 | - | - | 52.6/54.3 | - |
| LongVU-7B (Zhang et al., 2024d) | - | 67.6 | 65.4 | - | - | - | 60.6/- | 66.9 |
| LLaVA-OV-7B (Li et al., 2024a) | 56.6 | 60.1 | 64.7 | 79.4 | 57.1 | 56.5 | 58.2/61.5 | 56.7 |
| mPLUG-Owl3-8B (Ye et al., 2024) | - | - | - | 78.6 | - | 52.1 | 53.5/- | 54.5 |
| LLaVA-Video-7B (Zhang et al., 2024e) | 56.5 | 57.3 | 70.8 | 83.2 | 67.9 | 58.2 | 63.3/69.7 | 58.6 |
| Qwen2.5-VL-7B (Wang et al., 2024c) | - | - | - | 70.2 | 70.5 | 54.7 | 65.1/71.6 | 69.6 |
| InternVL2.5-8B (Zhang et al., 2024c) | - | 51.5 | 68.9 | - | - | 60.0 | 64.2/66.9 | 72.0 |
| InternVL2.5-8B-ITG-32 | 57.4 | 51.6 | 75.0 | 79.5 | 64.9 | 61.9 | 67.3/69.6 | 72.2 |

## C    DATASET DETAILS

### C.1    PROMPT TEMPLATE

Our Question-guided Clip Retrieval process utilizes a carefully designed prompt template (shown in Table 8) that instructs the LLM to analyze chronologically ordered clip-level descriptions and identify the minimal set of clips necessary to answer a given question. The prompt template consists of three main components:

- **Task Description**: Defines the LLM's role as an expert in analyzing video clip descriptions and establishes the goal of selecting clips that cover both question and answer content.

- **Guidelines**: Provides detailed instructions for clip selection, including handling time-related expressions, determining if a single or multiple clips are needed, addressing questions about object existence or movement, and avoiding unnecessary clips.

- **Output Format**: Specifies the required JSON structure for responses, ensuring consistent formatting with explanation and clip number fields.

This template enables the LLM to perform chain-of-thought reasoning when selecting relevant clips. The model analyzes keywords from questions, identifies temporal relationships (e.g., "before," "after"), and provides explicit rationales for its selections. For cases where no relevant clips exist, the model returns "None" to reduce annotation noise.

We implement this process using GPT-4o-mini (OpenAI, 2024), which is sufficient for accurate clip selection while reducing annotation costs by over 10 times compared to larger models. The selected clips are then converted to event boundaries defined by timestamps based on frame indices for the final temporal grounding annotations.

Table 7: Dataset quality (IoU). We evaluate the performance in both multiple-choice (MC) and open-ended (OE) questions.

| Method | Semantic-MC | Semantic & Motion-OE | Semantic-MC | Semantic & Motion-OE |
|---|---|---|---|---|
| Qwen2.5-VL-32B | 0.31 | 0.36 | 0.27 | 0.37 |
| GPT4o | 0.24 | 0.30 | 0.26 | 0.27 |
| Ours | **0.79** | **0.74** | **0.72** | **0.69** |

Table 8: Prompt Template: An expert system for temporal localization in video segments. The system analyzes video segment descriptions to determine the minimal and necessary combination of segments required to answer questions.

**Task:**
You are an expert in analyzing video clip descriptions. Your task is to select which clip or combination of clips is necessary to answer the given question, ensuring the selected clips effectively cover the content of both the question and the answer.

**Guidelines:**

- Carefully read the descriptions to determine which clip(s) provide relevant content for the question and the answer.

- Clip descriptions are in chronological order. Use clip number to locate clips based on time-related expressions (e.g., "at the beginning of the video" suggests a smaller clip number, while "at the end of the video" suggests a larger one).

- First, determine if one clip can answer the question or if multiple clips are needed. Then, return a list containing the selected clip(s) and an explanation.

- If the question asks about the existence/movement of an object or event. The object/action/movement may not exist, meaning you can't find the answer in the description, but the question might still provide some clues. You need to find the sentence closest to those clues.

- If asked about the whole video description or overall atmosphere, you should return all clip numbers.

- If multiple clips provide similar descriptions of the content and any of them can be used to answer the question, return all corresponding clips.

- If there are no clues in all descriptions and cannot answer the question, return "None.".

- **Important**: Avoid including unnecessary clips.

**Output Format:**
1. Your output should be formed in a JSON file.
2. Only return the Python dictionary string.
For example:
```
{"explanation":  "...", "clip_num":  "One clip:  [Clip-2]"}
{"explanation":  "...", "clip_num":  "Multiple clips:
[Clip-1, Clip-7, Clip-8]"}
{"explanation":  "...", "clip_num":  "None."}
```

## C.2 HUMAN-IN-THE-LOOP VERIFICATION

Ensuring the quality of automatically annotated datasets is critical for the reliability and effectiveness of downstream video understanding models. In this work, we implement a comprehensive quality control protocol for the VideoITG-40K dataset.

Our pipeline begins with diverse sampling: we select a representative subset of the dataset, covering a wide range of instructions and video scenarios. For this subset, we conduct human verification,

Table 9: Prompt template for identifying motion-related questions in video QA tasks. The template instructs the system to analyze each question-answer pair and determine whether the question pertains to absolute or relative speed, responding with "Yes" or "No" accordingly. Example cases are provided for clarification.

**Task:**
Analyze the given QA pair to determine if the question is related to speed. Specifically, check if it involves either absolute speed (the speed of a specific object) or relative speed (comparing the speed of different objects). Provide an output of "Yes" if the question pertains to speed, and "No" otherwise.
**Important**: Respond with "Yes" or "No" only.

**Example:**
**Question 1:** Which is faster, the white car or the bicycle? Options: A. The bicycle. B. The white car. C. Both are at the same speed. D. None of the above.
**Answer 1:** B. The white car.
**Output:** Yes.
**Question 2:** What color is the cat ?Options: A. black B. white C. orange D. gray
**Answer 2:** C. orange
**Output:** No.

Table 10: Prompt template for identifying semantic-related questions in video QA tasks. The template instructs the system to analyze each question-answer pair and determine whether the question pertains to absolute or relative speed, responding with "Yes" or "No" accordingly. Example cases are provided for clarification.

**Task:**
Analyze the given QA pair to determine if the question inquires about the existence of an object or action. If it does, and the answer is "No" (indicating non-existence), output "Yes." If the question is not about existence, or the answer is "Yes" (indicating existence), output "No."
**Important**: Respond with "Yes" or "No" only.

**Example:**
**Question 1:** After going through the bag, does the person meticulously clean the area around the sink?
**Answer 1:** No, the person does not clean the area around the sink after going through the bag. The video primarily focuses on the action of the person with the bag and items, not on cleaning activities.
**Output:** Yes.
**Question 2:** Is there a cat sitting on the windowsill in the video?
**Answer 2:** Yes, there is a cat sitting on the windowsill throughout the video.
**Output:** No.

where expert annotators review the automatically generated annotations to assess their accuracy and relevance. This process allows us to identify and correct potential errors, and to further calibrate our annotation pipeline for improved consistency and quality.

As shown in Table 7, we compare our pipeline with baselines where advanced models such as Qwen2.5VL and GPT-4o are directly prompted to answer the temporal boundaries of relevant events. These direct approaches result in significantly lower performance, highlighting the advantage of our multi-step, instruction-guided annotation strategy.

### C.3 FRAME SAMPLING ALGORITHM

The algorithm in 1 is designed for semantic-only keyframe selection, aiming to extract a diverse set of frames that comprehensively capture the semantic content of a video—such as people, scenes, or objects. By leveraging CLIP features, the algorithm compares each frame to previously selected keyframes using a bidirectional similarity measure. Frames are selected when their semantic features differ significantly from both the last keyframe (scene change threshold) and from future frames (diversity threshold), ensuring that each chosen frame represents distinct semantic information. This process produces a set of keyframes with maximal semantic coverage and minimal redundancy, aligning with the goal of representing all major semantic aspects of the video.

## D VISUALIZATION

In Fig. 5 and Fig. 6, we present two sets of results comparing sampling results of VideoITG with uniform sampling. Fig. 5 demonstrates a temporal reasoning problem, where our model accurately identifies the "workout" mentioned in the question and successfully locates the subsequent actions in the video, leading to the correct answer selection. In contrast, the uniform sampling strategy failed to capture these crucial frames. Fig. 6 illustrates a non-existence question scenario where our model effectively identifies all IMAX movies present in the given options, enabling it to successfully filter out and determine the correct answer.

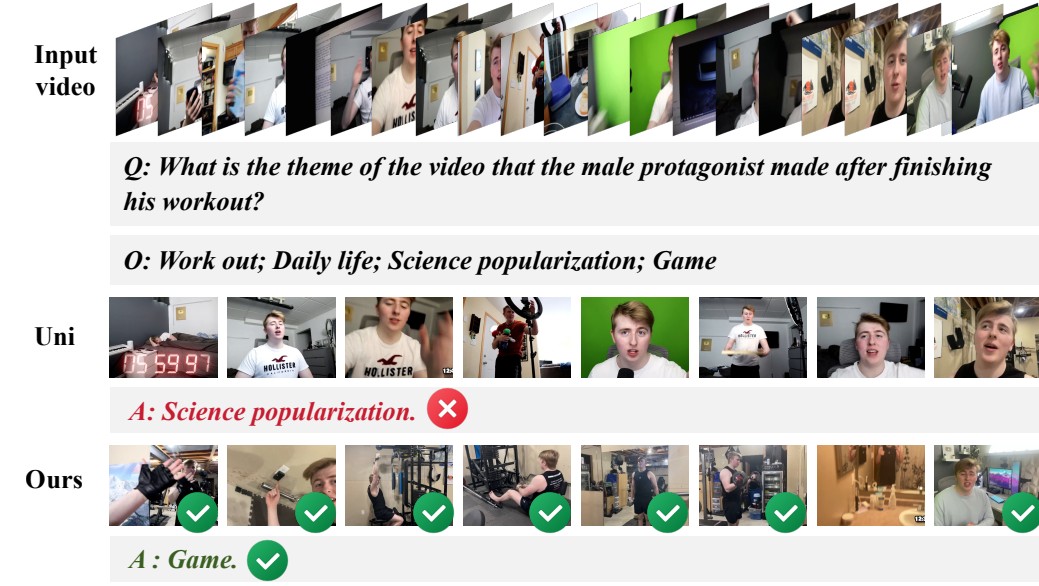

Figure 5: Example-1 shows how different sampling strategies impact video understanding. We mark the identified key frames that directly answer the question with green check-marks.

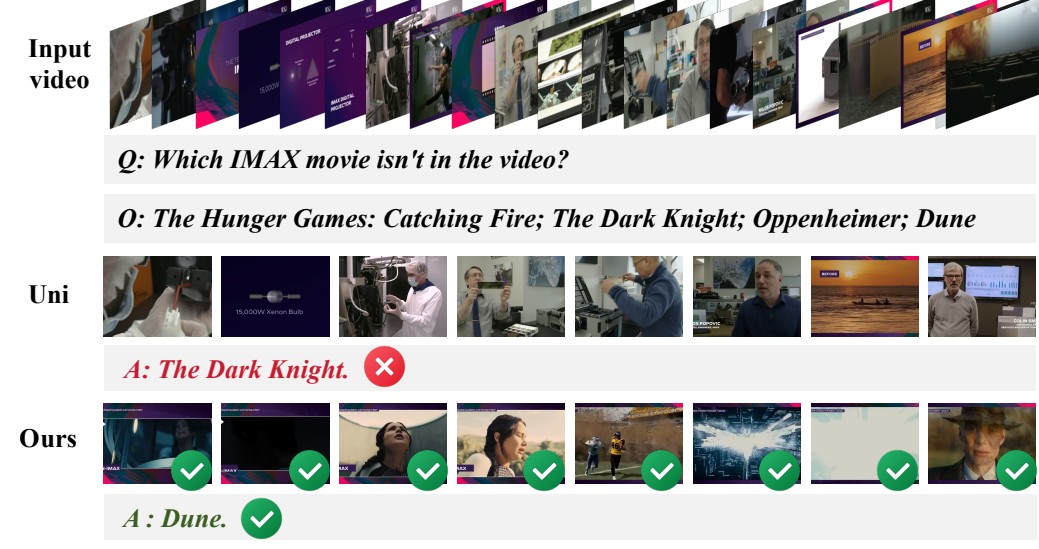

Figure 6: Example-2 shows how different sampling strategies impact video understanding. We mark the identified key frames that directly answer the question with green check-marks.

