# OpenReview forum: "VideoITG: Multimodal Video Understanding with Instructed Temporal Grounding"
_ICLR.cc/2026/Conference — ICLR 2026 Conference Withdrawn Submission_

### Official Review · Reviewer_HBJk · 2025-10-30

**Soundness:** 3
**Presentation:** 4
**Contribution:** 2
**Rating:** 4
**Confidence:** 4

**Summary:**

This paper introduces VIDEOITG, a novel framework for instruction-guided temporal grounding in video understanding. The key idea is to dynamically select informative frames based on user instructions, rather than relying on uniform or heuristic sampling. To achieve this, the authors propose VidThinker, an automated annotation pipeline that generates instruction-aligned temporal grounding labels, and construct VideoITG-40K, a large-scale dataset containing 40K videos with 500K instruction-based annotations. Furthermore, they design a plug-and-play model that integrates with existing Video-LLMs to enhance frame selection quality and downstream task performance. Extensive experiments demonstrate consistent improvements across multiple benchmarks, with particularly notable gains in long-form video understanding tasks.

**Strengths:**

1. The paper addresses a real and underexplored problem: how to select frames adaptively based on user instructions, which is more flexible and practical than static or uniform sampling.
2. VideoITG-40K is currently the largest instruction-guided temporal grounding dataset, and the automated annotation pipeline (VidThinker) is well-designed and interpretable.
3. The proposed method consistently improves performance across multiple benchmarks (e.g., +9.0% on CG-Bench, +8.6% on MLVU) when integrated with various Video-LLMs (e.g., InternVL2.5, LLaVA-Video).
4. The paper includes thorough ablations on model design choices, dataset construction, and pre-training strategies, validating the importance of each component.

**Weaknesses:**

1. While the task is novel, the model design (e.g., anchor tokens, pooling-based classification) appears to be incremental and shares similarities with existing vision-language modeling techniques. Although the framework effectively adapts current methods to the new task, it does not introduce significant architectural breakthroughs.
2. The paper primarily focuses on demonstrating the effectiveness of its dynamic frame selection approach but does not extensively compare it with some stronger frame selection baselines, such as reinforcement learning-based methods, temporal attention mechanisms, or query-aware sampling strategies.

**Questions:**

1. What is the additional inference latency introduced by VideoITG?
2. What is the accuracy of the labels generated by VideoITG model? Is there any quantitative or qualitative analysis?

---

### Official Review · Reviewer_NNgW · 2025-10-30

**Soundness:** 2
**Presentation:** 2
**Contribution:** 2
**Rating:** 4
**Confidence:** 4

**Summary:**

VideoITG is a novel framework that enhances multimodal video understanding by introducing Instructed Temporal Grounding—a task where frame selection is conditioned on user instructions. To support this, the authors propose VidThinker, an automated annotation pipeline that generates instruction-aligned temporal labels without human effort, and construct VideoITG-40K, a large-scale dataset with 40K videos and 500K instruction-grounded annotations. The plug-and-play VideoITG model uses vision-language alignment to select task-relevant frames, significantly improving performance across benchmarks like CG-Bench (+9.0%), MLVU (+8.6%), Video-MME (+4.0%), and LongVideoBench (+3.6%) when integrated with InternVL2.5-8B.

**Strengths:**

1. The paper introduces Instructed Temporal Grounding, a novel task formulation that conditions frame selection on free-form user instructions rather than on fixed heuristics or single textual queries.   This reframing turns frame sampling into a language-guided, task-adaptive decision problem.
2. The dataset construction is meticulous. VidThinker performs three cascaded checks (clip captioning, relevance retrieval, frame-level filtering) and integrates human-in-the-loop spot evaluations, yielding >0.7 IoU against expert annotation.
3. The plug-and-play nature of VideoITG means practitioners can retrofit existing models without architectural surgery.

**Weaknesses:**

1.The baseline model used in the article is too outdated; we would like to see results using internvl3, internvl3.5, or qwen2.5vl.
2. Limited instruction diversity and linguistic complexity. Although 500 k QA pairs are generated, the prompt templates (Appendix C.1–C.10) reveal that most questions are single-sentence, factoid-style, and temporally local (“What did X do before Y?”). There are no instructions that require multi-hop reasoning across ≥3 disjoint moments, no anaphoric references (“After he finished that …”), and no negation plus existential checks (“Show me a scene where the dog is not on the sofa”).

**Questions:**

Please refer to the weaknesses.

---

### Official Review · Reviewer_DQ1E · 2025-10-31

**Soundness:** 1
**Presentation:** 2
**Contribution:** 3
**Rating:** 2
**Confidence:** 4

**Summary:**

The paper introduces VideoITG, a framework for improving frame selection in long videos by aligning sampling with user instructions. It proposes an automated annotation pipeline, VidThinker, that uses LLMs to generate instruction-conditioned captions, retrieve relevant clips, and localize key frames. This process results in the VideoITG-40K dataset, consisting of 40K videos and 500K temporal grounding annotations. The authors then train a VideoITG model that selects frames based on instruction relevance before feeding them to Video-LLMs for question answering. Experimental results on multiple video understanding benchmarks (e.g., CG-Bench, MLVU, Video-MME, LongVideoBench) show consistent performance gains, demonstrating the potential of instruction-guided frame selection for improving multimodal reasoning in long videos.

**Strengths:**

The following are the strengths of the paper:

**1. Addresses an important and relevant problem.** The paper targets the challenge of selecting informative frames from long videos, which is a key bottleneck in Video-MLLMs. The topic is timely and valuable for improving the scalability of Video-MLLMs.

**2. Reasonable annotation and training pipeline.** The proposed VidThinker pipeline, combining clip captioning, retrieval, and frame localization, forms a systematic way to create grounding data and train a frame-selection model before feeding selected frames to an MLLM for question answering. This structure is logical and useful for constructing instruction-grounded datasets.

**3. Practical step toward efficient video understanding.** While the ideal scenario would have an MLLM process all 512 frames directly but efficiently, the proposed method’s selection of the most relevant 32/64 frames is a meaningful compromise, making the task slightly computationally feasible while preserving essential context. It represents a step forward for long-video reasoning.

**Weaknesses:**

The following are the weaknesses of the paper:

**1. Lack of clarity in instruction selection mechanism.** The paper does not explain how the system decides which instruction type (semantic, motion, both, or non-clues) applies to a given question-answer pair. It is unclear whether this is done by an LLM, heuristic rules, or predefined templates. Further, how this decision differs from simply using the original question?

**2. Noise accumulation through multiple stages.** Each step of annotation (phrase generation, clip retrieval, and frame localization) relies on automatic LLM or MLLM predictions. Without human verification or filtering, errors from earlier stages may propagate, leading to noisy or inconsistent annotations.

**3. Questionable need for the phrase generation step.** The LLM-generated intermediate phrases may introduce hypothetical details not present in the video (e.g., LLM generated “the man hits the drums with his hands”, but there is no information in the question or answer that the man hits the drums with his hands), causing hallucinations or bias rather than improving grounding accuracy.

**4. Unverified claim about annotation quality.** The claim that conditioning on instruction and answer cues ensures “relevant and informative” annotations (lines 187-188) is not supported by quantitative validation or human evaluation. No evidence is shown that the generated annotations are indeed high quality.

**5. Unexplained temporal reasoning mechanism.** The paper states that LLMs infer temporal relationships (lines 192–195) but provides no details on how this is achieved, which model is used, or what prompts enable this reasoning.

**6. Ambiguity between “LLM” and “MLLM” usage.** Lines 200-201 says “prompt a large language model (LLM) … given the QA pair and a single frame”. A text‑only LLM cannot view frames and an MLLM would be required. The paper uses “LLM” vs “MLLM” inconsistently, leaving a core part of the pipeline difficult to follow.

**7. Computational cost of Variant C model.** The paper's preferred model (Variant C) uses full attention, which would be computationally heavy when processing up to 512 frames as input, a practical limitation. How does the Variant C compares with Variants A & B in terms of computation cost (e.g. Inference Time)? This comparison, in addition to downstream performance, is required to justify design choice.

**8. Unfair experimental comparisons.** VideoITG used 512 frames at 1fps to pick 32, while baselines process only 32 uniform frames. This gives VideoITG more visual context, making the reported improvements difficult to interpret as fair comparisons. A fair experimental setting would be when the baseline MLLM also processes 512 frames. For example with Qwen2.5-VL which can process over 512 frames at native resolution.

**9. Unsupported claims.** Statements about CoT prompting benefits (lines 195-197) or “high precision” frame selection (lines 206-207) are not supported by ablation studies or controlled experiments.

**10. Ambiguity in ablation details.** In ablations (Table 3), it’s unclear whether “7B” and “3B” refer to the backbone MLLM or the VideoITG model. Which baseline model is used in the ablations? Relating it with Table 1, it seems to be InternVL2.5-8B, but it is not explicitly mentioned.

**11. Lack of explanation for ablations.** In Table 3, when removing steps such as “Clip Captioning” or “Frame Localization,” the paper does not describe how the pipeline functions without them, (e.g., how clips or key frames are selected in these cases).

**12. Incomplete human-verification.** Although a “human-in-the-loop” verification is mentioned in the appendix, no details on sample size, annotation consistency, or inter-annotator agreement are provided, leaving annotation reliability uncertain.

---

Despite addressing a relevant problem, the paper suffers from unclear methodological details, possible noise accumulation, and unverified claims about annotation quality. Its evaluation lacks fairness, leaving the contribution technically incomplete. Overall, while promising in idea, the work lacks critical validation and methodological transparency to justify acceptance at a top conference.

**Questions:**

Please refer to the *Weaknesses* section for detailed clarifications required from the authors.

---

### Official Review · Reviewer_rBb9 · 2025-11-11

**Soundness:** 1
**Presentation:** 3
**Contribution:** 2
**Rating:** 2
**Confidence:** 3

**Summary:**

The paper introduces VideoITG, a plug-and-play framework that injects instruction-guided temporal grounding into frame selection for Video-LLMs. An automated pipeline VidThinker constructs VideoITG-40K (40K videos, 500K annotations) by instruction-conditioned captioning, LLM-scored clip retrieval, and frame-level Yes/No filtering. The authors explore three designs (Fig.3): Variant-A (text generation), Variant-B (anchor-based classification under causal attention), and Variant-C (pooling-based classification under full attention). They report consistent gains over uniform sampling across LongVideoBench, MLVU, VideoMME, CG-Bench with multiple backbones.

**Strengths:**

1. Architectural exploration is meaningful: Fig.3 compares three alternatives under realistic constraints—text generation vs. discriminative classification, causal vs. full attention—and the ablation (Table 3) supports the preference for Variant-C. This is a useful design-space study for practitioners.


2. Data pipeline is systematic: VidThinker’s three-stage process (caption, then retrieval, and frame-level filtering) gives a coherent, interpretable way to obtain instruction-aligned labels at scale, including a sensible taxonomy of instruction types with matching sampling strategies (Fig.1–2).

3. Reported gains across backbones: Uniform-vs-ITG Top-k comparisons show consistent improvements on multiple VLMs (e.g., +5.6 avg for InternVL2.5-8B; +5.1 avg for InternVL2.5-26B).

**Weaknesses:**

1. **Task definition is confusing**: The paper toggles between “temporal grounding” and “temporal localization”, and positions VideoITG as “instruction-driven temporal grounding.” However, the formal definition and evaluation protocol are not crisply specified (what exactly is a correct grounding under multi-clue instructions? how do we measure frame-level correctness vs. downstream QA?). The intro text mentions general grounding vs. localization, but an unambiguous, task-level definition and metrics are missing in the main method section.

2. **Related-work coverage is incomplete**: Given the recent surge of instruction-tuned long-video datasets and selection methods, the paper’s related-work section does not convincingly situate VideoITG among instruction-centric datasets (e.g., GenS-150K[1], which is not discussed) nor provide head-to-head results on more closely aligned resources. The bibliography lists many long-video works, but the core line of instruction-conditioned frame retrieval/sampling methods, such as works[1, 2,3], is under-analyzed; this weakens the novelty and fairness of positioning.

[1] Generative Frame Sampler for Long Video Understanding. (ACL 2025)

[2] Adaptive Keyframe Sampling for Long Video Understanding. (CVPR 2025)

[3] Frame-Voyager: Learning to Query Frames for Video Large Language Models. (ICLR 2025)

3. The paper **does not report the additional inference latency** introduced by frame retrieval / grounding.

The method requires scanning 512 frames to score relevance. Appendix B only reports absolute inference time (≈6.42s end-to-end, with ≈0.61s due to VideoITG), but never compares this overhead to uniform sampling or to other frame selection methods.

**Questions:**

Refer to the weakness.

---

### Note · Authors · 2025-11-12

I have read and agree with the venue's withdrawal policy on behalf of myself and my co-authors.